# Spatial and Temporal Variability of Saxitoxin-Producing Cyanobacteria in U.S. Urban Lakes

**DOI:** 10.3390/toxins16020070

**Published:** 2024-02-01

**Authors:** Youchul Jeon, Ian Struewing, Kyle McIntosh, Marcie Tidd, Laura Webb, Hodon Ryu, Heath Mash, Jingrang Lu

**Affiliations:** 1Oak Ridge Institute for Science and Education, Oak Ridge, TN 37830, USA; 2United States Environmental Protection Agency, Office of Research and Development, Cincinnati, OH 45268, USA; 3United States Environmental Protection Agency, Region 8, Lakewood, CO 80225, USA; 4United States Environmental Protection Agency, Region 7, Kansas City, KS 66101, USA

**Keywords:** paralytic shellfish toxins, saxitoxin, *sxtA*, harmful cyanobacterial blooms, cyanobacteria, qPCR

## Abstract

Harmful cyanobacterial blooms (HCBs) are of growing global concern due to their production of toxic compounds, which threaten ecosystems and human health. Saxitoxins (STXs), commonly known as paralytic shellfish poison, are a neurotoxic alkaloid produced by some cyanobacteria. Although many field studies indicate a widespread distribution of STX, it is understudied relative to other cyanotoxins such as microcystins (MCs). In this study, we assessed eleven U.S. urban lakes using qPCR, *sxtA* gene-targeting sequencing, and 16S rRNA gene sequencing to understand the spatio-temporal variations in cyanobacteria and their potential role in STX production. During the blooms, qPCR analysis confirmed the presence of the STX-encoding gene *sxtA* at all lakes. In particular, the abundance of the *sxtA* gene had a strong positive correlation with STX concentrations in Big 11 Lake in Kansas City, which was also the site with the highest quantified STX concentration. Sequencing analysis revealed that potential STX producers, such as *Aphanizomenon*, *Dolichospermum*, and *Raphidiopsis,* were present. Further analysis targeting amplicons of the *sxtA* gene identified that *Aphanizomenon* and/or *Dolichospermum* are the primary STX producer, showing a significant correlation with *sxtA* gene abundances and STX concentrations. In addition, *Aphanizomenon* was associated with environmental factors, such as conductivity, sulfate, and orthophosphate, whereas *Dolichospermum* was correlated with temperature and pH. Overall, the results herein enhance our understanding of the STX-producing cyanobacteria and aid in developing strategies to control HCBs.

## 1. Introduction

In the past few decades, harmful cyanobacterial blooms (HCBs) have become an increasing global threat to human and environmental health. HCBs can lead to the deterioration of aquatic environments, such as lakes and rivers, by depleting dissolved oxygen and producing cyanotoxins [1,2]. HCB events involving the production of cyanotoxins, such as microcystins (MCs), cylindrospermopsins (CYNs), and saxitoxins (STXs), can induce acute and chronic toxicity in fish, marine mammals, wild terrestrial animals, and humans [1,2].

Paralytic shellfish poisoning (PSP) by consuming paralytic shellfish toxins is considered to be a significant poisoning syndrome [3]. The poisoning effects (e.g., vomiting, burning sensation in the oral cavity, and facial numbness) can occur quickly after ingestion, and if proper measures are not taken in time, intoxicated individuals may lose their lives due to respiratory arrest or cardiovascular shock [4,5]. The category of toxins that triggers PSP comprises STXs and an array of over 50 structurally related analogues [6]. These toxins are produced by some marine dinoflagellates such as *Alexandrium *spp. and *Pyrodinium bahamense* [7]. In 1995, it was found that the freshwater cyanobacterium *Dolichospermum circinalis* (formerly *Anabaena circinale*) produced the toxin and, since then, other freshwater genera such as *Raphidiopsis* and *Aphanizomenon* have been reported to produce STX, including the variants [8,9].

A qPCR technique targeting genes associated with cyanotoxin synthesis is a promising approach to cost-effectively monitor the potential for cyanotoxin production. This technique could offer the detection and quantification of STX producers for the characterization and prediction of HCBs in environments, particularly in aquatic ecosystems. In a previous study, the *sxt* gene cluster from a filamentous cyanobacterial strain *Raphidiopsis raciborskii* T3 was first identified to be closely involved in the biosynthetic pathway of STXs [10]. Since the size of the *sxt* gene cluster varies among different species, ranging from 25.7 kb to 35 kb, specific *sxt* genes, such as *sxtA, sxtB*, and *sxtG,* have been targeted to identify and estimate the abundance of STX-producing microorganisms [11,12]. For instance, a study reported that 8 out of 29 strains of *Dolichospermum* from 12 regions of Australia detected the *sxtA*, *sxtL*, *sxtN*, and *sxtSUL* genes through qPCR, whereas the *sxtX* gene was not detected in the strains [13]. In addition, a newly designed primer set targeting the *sxt* gene was tested with water samples from five Danish lakes potentially containing dominant STX producers, such as *Aphanizomenon*, *Dolichospermum*, and *Planktolyngbya* [14]. The authors observed a significant positive correlation (regression *R*^2^ = 0.64, *p*-values < 0.01) between *sxtA* copy number and total STXs. Despite recent progress in this area, most of the cyanobacterial qPCR research focuses on genes related to MC producers or STX producers of eukaryotic marine dinoflagellates. For this reason, freshwater cyanobacteria are still underrepresented in this area.

Urban lakes play a crucial role in providing essential services such as storm water retention to prevent flooding and pollutant load mitigation [15,16]. They frequently serve as a source of potable water and are used for recreational activities [17]. However, the urban lakes typically characterized by their small size, shallow depth, and limited water circulation are prone to eutrophication. Conditions such as high temperatures and alkalinity further enhance this susceptibility, providing an ideal environment for the occurrence of HCBs [18,19]. Given the potential exposure to water contaminated with cyanobacteria, it becomes imperative to assess the associated risks.

In this study, we aimed to enhance the understanding of STX-producing cyanobacteria dynamics in urban lakes and, specifically, to determine whether the existence of the *sxtA* gene is indicative of STX production and if there is a correlation between the copy number of the *sxtA* gene and toxin concentration. Moreover, we examined the relationships between STX concentrations, the abundance of identified STX producers, and important environmental parameters.

## 2. Results

### 2.1. Microbial Community Structures and Potential STX Producers

To understand the biological environment, the community structures of cyanobacteria and their associated bacteria were characterized. In 2021, samples were collected from 11 urban lakes in various locations and dates in three cities (Figure 1A, Appendix A). Upon examining the bacterial taxa present in the lake dataset, it was found that the community structure primarily consisted of five different phyla: Bacteroidota (30.6 ± 11.4%), Proteobacteria (29.5 ± 9.2%), cyanobacteria (20.7 ± 17.0%), Actinobacteriota (12.5 ± 7.8%), Verrucomicrobiota (5.3 ± 5.1%), and Planctomycetota (3.0 ± 2.5%) (Figure 1A). The relative abundance of Cyanobacteria varied across different sites, showing an increasing trend from July to August in most lakes. For Big 11 Lake and Tomahawk Creek Lake, their relative abundance continued to increase in November. Cyanobacteria exhibited a high relative abundance in July at South Lake and Sloans Lake, decreased in August, and peaked again in November at South Lake. In Figure 1B, alpha diversity indices (Faith and Pilou), reflecting microbial community diversity and evenness, were calculated. The results showed that the communities of the lakes in Cincinnati were more diverse and evenly distributed than those in Kansas City and Denver. It is worth mentioning that there was no significant difference in alpha diversity between the lakes in Denver and Kansas City. However, the lakes in Cincinnati exhibited a significantly higher alpha diversity than the lakes in the other two cities (Kruskal Wallis, *p* < 0.05). In Kansas City, the samples from Big 11 Lake, South Lake, and Tomahawk Creek Lake had relatively low alpha diversity indices. To evaluate dissimilarities in the overall microbial community composition across the various lakes, PERMANOVA, clustering analysis, and NMDS ordination were employed. The results of the PERMANOVA revealed notable differences in microbial communities among the lakes, depending on the city in which they were located (*F* = 8.2, *p* < 0.001). Additionally, the NMDS plot exhibited that lake samples collected from Cincinnati were predominantly grouped on the left (Figure 1C). In contrast, samples in Kansas City and Denver were scattered horizontally, while some of them in Kansas City and Denver exhibiting a significant abundance of Cyanobacteria (e.g., Big 11, Tomahawk Creek, and Sloans Lakes) were grouped on the upper- and lower-right sides, forming dispersed clusters.

At the genus level, the community composition results displayed distinct bloom characteristics that were unique to each region (Figure 2). In Kansas City samples, *Aphanizomenon* and *Planktothrix* were dominant, with *Planktothrix* being the overwhelmingly dominant genus at peak in Big 11 Lake (68.4%), South Lake (59.6%), and Tomahawk Creek Lake (65.2%). In Chaumiere Lake, *Dolichospermum* was the dominant cyanobacterial species, with a relative abundance peak of 21.7%. For Lake of the Wood, Cyanobacteria accounted for only 2.5% of the total abundance at the peak in July. Meanwhile, samples from the lakes in Cincinnati were mainly composed of *Cyanobium* and *Microcystis*, but their overall relative abundance was low, with 10% at peak. Similarly, samples from the lakes in Denver also showed similar taxa compositions, with low relative abundances of Cyanobacteria, but Sloans Lake was mainly composed of *Dolichospermum* and *Aphanizomenon* and showed a decreasing trend over time.

### 2.2. Temporal Variations in sxtA Gene and STX Production

The qPCR analysis showed that *sxtA* genotypes were present in all lakes (Figure 3). Compared to the other lakes, Tomahawk Creek Lake exhibited a significantly higher mean abundance of the *sxtA* gene (3.38 log_10_GCN∙mL^−1^). In addition, the highest level of the *sxtA* gene was observed at Chaumiere Lake in early July (4.27 log_10_GCN∙mL^−1^). The average abundance of the *sxtA* gene detected in the Kansas City (2.84 log_10_GCN∙mL^−1^) and Denver areas (2.41 log_10_GCN∙mL^−1^) was significantly higher than that detected in the Cincinnati area (1.69 log_10_GCN∙mL^−1^).

The temporal trends in STX production, which were similar to the trends for the abundances of the *sxtA* genes, varied from 0 to 0.913 μg∙L^−1^ at the sampling sites (Figure 3). The highest concentration of STX was detected in Big 11 Lake in Kansas City. STXs were also found in the other four lakes in Kansas City, with concentrations ranging from 0 to 0.236 μg∙L^−1^. In Sloans Lake near Denver, STXs were observed in the range of 0 to 0.067 μg∙L^−1^. Despite the presence of the *sxtA* gene in all of the lakes in Cincinnati, STXs were undetectable (less than 0.05 μg∙L^−1^).

To better understand the potential for STX production in urban lakes, high-depth sequencing on amplicons from the *sxtA* gene was performed. These data can help confirm various taxa to produce STX and provide insight into the distribution and diversity of the *sxtA* gene. It is important to note that the primers used for the qPCR assay of the *sxtA* gene are different from the one used for target sequencing (Appendix A), but both primers mainly target the *sxtA* gene in *Aphanizomenon* and *Dolichospermum* [20,21]. The analysis revealed that a single ASV was dominantly present in the urban lakes, and the BLAST search against the NCBI database (cut-off at 80%, identity at 70% E-value at 10^−7^) confirmed that it was associated with *Aphanizomenon*, *Dolichospermum*, *Heteroscytonema*, and *Lyngbya* (Table 1). These genera were also detected in our 16S rRNA amplicon sequencing analysis. Kansas City lakes showed more ASVs than Denver and Cincinnati, with Big 11 Lake having the most from *sxtA* gene sequencing (Figure 3). The distributions of the ASV showed a similarity to the trend observed in the abundances of the *sxtA* genes obtained by qPCR.

### 2.3. Correlation between sxtA Abundance and PCR Measurements to STX Production

Based on the qPCR and sequencing results in this study, Pearson correlation tests were performed, and potential STX producers in the lakes were examined (Table 2 and Figure 4). The correlation between *sxtA* gene abundances by qPCR and STX concentrations was significant only in the Big 11 Lake, where STXs were detected, with a correlation coefficient of 0.69 (*p*-values < 0.05). In Chaumiere Lake and Sloans Lake, the *sxtA* gene abundances negatively correlated with the STX concentration (*R*_pearson_ −0.42 and *R*_pearson_ −0.61, *p*-values > 0.05). The correlations between the abundances of the *sxtA* gene and the toxin level in South Lake and Tomahawk Creek Lake were positive. However, their significance was low (*R*_pearson_ 0.3 and *R*_pearson_ 0.36, *p*-values > 0.05). Similarly, when comparing the qPCR-based abundances of the *sxtA* gene to the abundances of the amplified ASV obtained through deep sequencing, no significant correlation was observed in the lakes, except for Sloans Lake in Denver (*R*_pearson_ 0.9, *p*-values < 0.05). South Lake and Campbell Lake exhibited high correlation coefficients, but the results were not statistically significant (*R*_pearson_ 0.69 and *R*_pearson_ 0.62, *p*-values > 0.05). In Figure 4, correlation coefficients were calculated to further explore the role of the cyanobacterial community in STX production by analyzing the relative abundance of each genus and qPCR results. The results showed that among the potential STX producers, the relative abundances of *Aphanizomenon* in Big 11 Lake were significantly correlated with *sxtA* gene abundances and STX concentrations (*R*_pearson_ 0.63 and *R*_pearson_ 0.65, *p*-values < 0.05). In Chaumiere Lake, *Aphanizomenon* was strongly correlated with *sxtA* gene abundances (*R*_pearson_ 0.72, *p*-values < 0.05). In contrast, *Dolichospermum* was found to be positively correlated with both parameters in Lake of the Wood (*R*_pearson_ 0.76 and *R*_pearson_ 0.84, *p*-values < 0.05).

### 2.4. Effects of Physicochemical Water Quality Parameters on STX Production

Previous studies have shown the relationship between physicochemical water parameters (e.g., nutrient levels, pH, and temperature) and factors relating to cyanotoxin production [22,23]. In this study, correlation analyses were conducted to investigate whether there was a relationship between environmental variables and either STX production or STX producers (Figure 5, Appendix A). The results revealed that there was no clear correlation between the environmental variables and STX production in our study (Figure 5A). Canonical correspondence analysis (CCA) was performed to assess the impact of various environmental and physiochemical variables from each lake on the differences in cyanobacterial taxa abundance. CCA showed that *Aphanizomenon* and *Raphidiopsis* corresponded with increasing cell count, conductivity, sulfate concentrations, PC, and orthophosphate as phosphorus (P) (Figure 5B). In the case of *Dolichospermum*, it showed correspondence with PC, PC:CHL, and pH. In addition, the relationships between environmental variables and the composition of cyanobacteria, specifically potential STX-producing genera, were evaluated by Pearson correlation testing (Figure 5C). *Aphanizomenon* strongly correlates with orthophosphate as *p*, sulfate, PC, and conductivity (*R*_pearson_ 0.42 to 0.56, *p*-values < 0.01) among the factors shown in the CCA results. *Dolichospermum* and *Cylindrospermopsis* also showed a significant relationship between PC (*R*_pearson_ 0.29 and 0.36, *p*-values < 0.05) and the PC:CL ratio (*R*_pearson_ 0.31 and 0.33, *p*-value < 0.05). On the other hand, Planktothrix was strongly correlated with the PC:CH ratio and cell count (*R*_pearson_ 0.42 to 0.53, *p*-values < 0.01) but had a negative correlation with temperature (*R*_pearson_ −0.54, *p*-value < 0.01).

## 3. Discussion

In this study, samples collected from eleven urban lakes showed that the relative abundances of cyanobacteria varied temporally and spatially. *Aphanizomenon* and *Planktothrix* mostly dominated five lakes in Kansas City (Figure 2). In comparison, four lakes in Cincinnati were dominant with *Cyanobium,* and *Microcystis* and two in Denver were composed of *Dolichospermum* and *Aphanizomenon*. In the lakes of Kansas City, the dominance of *Planktothrix* may be due to its efficient utilization of light at similar intensities, a key factor driving phytoplankton growth, and its adaptability to a wider range of temperatures compared to other cyanobacteria, especially *Dolichospermum* and *Aphanizomenon* [24,25,26]. In addition, it was found that the growth of *Aphanizomenon* is inhibited due to the allelopathic effect produced by *Microcystis* [27]. Recently, a study found that the growth inhibition of *Aphanizomenon* by *Microcystis* is unlikely to result from a single metabolite [28]. Instead, multiple metabolites, such as sphingolipids, glycerolipids, and succinylacetone, are involved in inducing the adverse phenotype. Similarly, it is speculated that the potential suppressive effect of *Planktothrix* could contribute to its dominance in lakes by inhibiting the growth of coexisting competitors [29]. To fully understand the competitive dynamics of *Planktothrix*, particularly at the onset of HCBs, further research focusing on the inhibitory influence exerted by *Planktothrix*, as well as the interplay of light and temperature conditions, is needed.

While the qPCR target, *sxtA,* covers cyanobacteria-producing STXs (e.g., *Raphidiopsis* and *Aphanizomenon*), the identified ASV has shown close relationships with *Aphanizomenon* and *Dolichospermum* (Table 1). The lakes in Kansas City exhibited a higher presence of the ASV compared to the lakes in Denver and Cincinnati. Notably, Big 11 Lake in July exhibited the highest number of ASVs from the *sxtA* gene-targeting sequencing. Considering that *Aphanizomenon* has a strong correlation with STX concentrations and the *sxtA* gene, as depicted in Figure 4, all evidence suggests that *Aphanizomenon* plays a significant role in the production of STXs in the lake. However, this correlation was not observed in the other lakes. Despite the increased prevalence of cyanobacteria genera other than *Aphanizomenon* and *Dolichospermum*, their correlations with the *sxtA* gene and STX concentration were not significant. For example, in South Lake and Tomahawk Creek Lake, *Raphidiopsis* was more prevalent compared to *Aphanizomenon* and *Dolichospermum*, but its correlation with STX concentration and the abundance of the *sxtA* gene was not significant. Consequently, it raises the possibility that *Raphidiopsis* might contribute to STX production through an alternative pathway not involving *sxt*-related genes, or it might be present without actively contributing, possibly due to some crucial steps missing within the *sxt* gene cluster [30,31].

Previously, qPCR signals from cyanotoxin genes, such as *mcy*, were used to forecast cyanotoxin levels and characterize the dynamics of toxin-producing cyanobacteria during HCBs. For instance, in the 2015 HCB at William H. Harsha Lake (Clermont County, OH, USA), qPCR and RT-qPCR signals of *mcy* were successfully established to predict whether total MC would exceed health advisory limits [32]. In this study, HCBs caused by STX-producing cyanobacteria in urban lakes, employing the qPCR assay based on the *sxtA* gene, were assessed. It is important to highlight that concentrations of MC observed during HCB are generally high, ranging from 1 µg/L to over 100 µg/L, in contrast to the comparatively lower concentrations of STX [33,34]. Despite this discrepancy, a strong correlation between qPCR and STX concentration was observed in Big 11 Lake, which recorded the highest concentration of 0.85 µg/L. However, results from lakes other than Big 11 Lake showed negative or low correlation coefficients (Table 2). This can be attributed to several factors. Firstly, the low concentration of STX and the lack of significant changes during HCBs make it difficult to observe a strong relationship with the abundance of the *sxtA* gene. Additionally, the weak qPCR signals from the assay, particularly when the abundances of the *sxtA* gene are low, can also interfere with the correlation between them. For instance, in most lake samples from Cincinnati, PCR amplification of the *sxtA* gene failed or was not high enough to be sequenced, even though most samples were positive with qPCR signals. Since the major genera of cyanobacteria identified in the Cincinnati samples were mainly *Cyanobium* and *Microcystis*, which are not known to produce STXs, it is plausible that the amplified ASV originated from certain non-STX-producing species harboring the *sxtA* gene. In addition, the absence of STXs in the lakes suggests that the gene is either not transcribed into mRNA or serves a different function unrelated to STX production, such as regulating cellular metabolism or synthesizing secondary metabolites [35,36]. For example, the *Scrippsiella trochoidea* dinoflagellate, which does not produce STX, was found to have 113 transcripts identified as homologous to *sxt* genes. These transcripts encompass 17 of the 34 genes present in the *sxt* genes of *Raphidiopsis raciborskii*, including the short isoform of *sxtA*1–*A*3. Hence, even when cyanotoxin-producing genes are present in abundance, their expression may be low, resulting in limited cyanotoxin production [37]. In order to obtain a more comprehensive understanding of STX production during a bloom, future monitoring efforts should incorporate the evaluation of transcript levels associated with the *sxtA* gene.

The presumption in this study is that the production of STX and the presence of the *sxtA* gene are not influenced by a singular environmental variable. Rather, it is likely that they are impacted by an intricate interplay of various factors. STXs can be continuously degraded by natural elements such as bacterioplankton or sunlight in the lake, while several factors, including nutrient input and temperature, can promote the production of cyanotoxins during HCBs. A recent study highlighted the ability of these natural factors to degrade STXs in eutrophic lakes [38,39]. The study found that lake bacterioplankton could decrease STX concentrations by 41–59%, a reduction also observed for four saxitoxin analogs. Furthermore, exposure to natural sunlight for 4–8 h reduced intracellular STXs by 38–52% while simultaneously increasing extracellular dissolved STXs by 7–29%. In line with our initial hypothesis, our findings revealed no substantial correlation between STX or the *sxtA* gene and any individual parameter (Figure 5). Instead, we did identify several abiotic parameters that exhibited a close association with *Aphanizomenon*, which is recognized as a potential producer of STX. However, it is important to note that biotic parameters, such as algicidal bacteria and cyanophages, also significantly influence the activity and population dynamics of cyanobacteria [40,41]. Among these parameters, orthophosphate as a P source stood out, demonstrating the highest correlation coefficient. In freshwater systems, P has traditionally limited primary production [42,43]. Under oligotrophic conditions, a scarcity of P tends to enhance STX production, potentially as an adaptive survival strategy in low-P environments [30,44]. However, the exact mechanisms involved are yet to be identified. A previous study demonstrated that elevated P levels, resulting from nutrient loading in Sandusky and Maumee Bays within western Lake Erie (OH, USA), lead to an increase in the relative abundance of *Aphanizomenon* [45]. This aligns with observations from Big Eleven Lake and Sloans Lake, where orthophosphate concentrations exhibited an upward trend. Specifically, concentrations ranged from 131 µg/L to 194 µg/L in Big Eleven Lake and from 157 µg/L to 303 µg/L in Sloans Lake, respectively (Appendix A). This abundance may be because communities shift towards being dominated by cyanobacteria, especially diazotrophs that can fix N, to meet their N needs when P loads increase.

## 4. Conclusions

Temporal and spatial variations in cyanobacteria abundances were monitored across eleven urban lakes. The qPCR analysis revealed that the *sxtA* gene targeting STX producers occurred at all lakes, even though STXs were only detected in the lakes from Kansas City and Denver. Big 11 Lake in Kansas City, where the highest concentration of STX was detected, had a significant correlation with the abundance of the *sxtA* gene (Rpearson 0.69). Amplicon-based sequencing targeting the *sxtA* gene showed that a single ASV was dominant across the lake sample, and it was taxonomically assigned to the genera Aphanizomenon and Dolichospermum. Given that, our results showed strong positive correlations between the relative abundances of Aphanizomenon and both *sxtA* gene abundances and STXs.

Due to the variability in samples along with lake types, the correlation analysis had limitations in revealing significant associations. Nevertheless, the analysis results suggest that the environmental factors, such as orthophosphate, sulfate, PC, and conductivity, corresponded with identified STX-producing taxa. Notably, Aphanizomenon has a strong correlation with orthophosphate. As the occurrence of STX-producing species continues to expand geographically, the insights gained from this study can be valuable in understanding STX production and accurately quantifying STX producers for more effective monitoring of HCB-associated adverse effects.

## 5. Materials and Methods

### 5.1. Study Sites and Sample Collection

Sampling sites at eleven lakes near three U.S. metropolitan areas (Kansas City, MO, USA; Cincinnati, OH, USA; and Denver, CO, USA) were chosen based upon having greater than 50% of urban land use in the surrounding area. Sites with the potential for water contact through recreational and angling activities were also considered. The sampling focused on the summer bloom period from June to September in 2021, with additional sampling events prior to bloom development in the spring and following bloom die-off in the fall. The names of the lakes and geographic coordinates of the sampling sites are shown in Appendix A.

### 5.2. Sample Collection and Measurements of Physicochemical Water Quality Parameters

Sampling was conducted weekly at the same location for each water body. Each sampling event consisted of in situ monitoring, passive sampler deployment, and grab sampling. Teams in each metropolitan area sampled within one day of each other. Samples were taken from shore or from fishing platforms in the top 0.5 m (photic zone) of the water column via stainless-steel pitchers. At each sampling event, a sample of approximately 3.5 L of water was collected using a stainless-steel cup attached to the end of a sampling pole. Several cups of water were used to fill a pitcher with the sample, which was then mixed and distributed into sample containers. Each cup was collected from approximately the same location and within 0.5 m of the lake’s surface. After collection, samples were split among coolers with ice and shipped on the same day to ensure that they arrived at the destination laboratory the following day. Filtration (pore size 0.45 μm; MilliPore; Foster City, CA, USA) occurred in the field for dissolved orthophosphate and in the laboratory for qPCR analysis.

Water quality meters (EXO1 Multiparameter Sonde or ProDSS Multiparameter, YSI Inc., USA) were utilized to measure physicochemical water quality parameters in situ. These parameters included temperature, dissolved oxygen, pH, conductivity, turbidity, and phycocyanin (PC). Meters were calibrated locally prior to each event according to in-house SOPs and manufacturer’s directions.

Chlorophyll *a* (CHL) was determined using EPA Method 445.0. Cell count was determined along with cyanobacterial identification utilizing FlowCam 8400 (Yokogawa Fluid Imaging Technologies, Scarborough, ME, USA). In addition, the counting chamber and light microscopy were utilized to count filamentous cells. Alkalinity was determined via Standard Method 2320B-1997. Anions, including chloride and sulfate, were analyzed according to EPA Method 300.0. Nitrate and nitrite were frozen upon arrival and analyzed according to EPA Method 353.2 and EPA Method 353.4, respectively. Dissolved orthophosphate as P was also frozen, then analyzed per EPA Method 365.5. For these analyses, a Lachat QC-8500 Flow Injection Autoanlyzer (Lachat Instruments, Loveland, CO, USA) and a Formacs TOC/TN analyzer (Skalar, Breda, The Netherlands) were used. For STXs, total toxin concentrations were determined following a triple freeze–thaw cycle to lyse cells, and an enzyme-linked immunosorbent assay (ELISA) was utilized by following the manufacturer’s procedure. Briefly, 50 µL of each standard (0.15 to 5 μg/L) and 50 µL of sample were dispensed into the ELISA wells, along with 50 µL of the antibody solution. The plate was then covered, gently swirled for 30 s, and allowed to incubate in the dark for 90 min. The contents were then poured out, and the plate was washed three times with 250 µL of wash buffer. Each well was filled with 100 µL of enzyme conjugate solution and left to incubate for 30 min. The plate was washed again with the wash buffer. Then, 100 µL of substrate color solution was added to each well. The plate was swirled for 30 s, incubated in the dark for 25 min, and, finally, 50 µL of stop solution was added. The plate was read at a wavelength of 450 nm using a Synergy H1 microplate reader, and the data were analyzed using a 4-parameter standard curve [46].

### 5.3. DNA Extraction and qPCR

Collected samples were filtered using a 0.45 polycarbonate µm membrane filter (Pall Corporation, Port Washington, NY, USA). The filters with captured biomass were stored in Lysing Matrix A tubes (MP Biomedicals, Irvine, CA, USA) with a cell lysis buffer (600 µL) and RNase inhibitor (QIAGEN, San Diego, CA, USA). The mixtures were stored at −80 °C before DNA extraction. Cell lysis was conducted for 1 min using a Mini-Beadbeater-16 (BioSpec Products, Inc., Bartlesville, OK, USA). After shaking, the tubes were centrifuged at 10,000× *g* for 3 min at room temperature, and the supernatant was transferred to a new tube. DNA in the supernatant was purified using the AllPrep DNA/RNA Mini Kit (QIAGEN). The concentration of purified DNA was measured by a Qubit^®^ 2.0 Fluorometer (Thermofisher, Waltham, MA, USA) and stored at −20 °C for future analysis.

To determine the abundances in the lake samples of the *sxtA* gene that encodes STXs, SYBR^®^ Green was used to perform qPCR on a QuantStudio™ 6 Flex System (Life Technologies Co., Carlsbad, CA, USA) [20,32,47,48]. It should be noted that general cyanobacterial STX producers with *sxtA* gene were targeted (Appendix A). Each reaction was composed of 10 μL of 2X SYBR^®^ Green Master Mix (Life Technologies Co.), 0.25 μM of primers (Integrated DNA Technologies, Inc., Coralville, IA, USA), 2 µL of 1 mg/mL BSA, and 2 μL of template DNA in a total reaction volume of 20 μL. The following thermal cycling conditions were applied: 40 cycles of 95 °C for 15 s, annealing temperatures of 60 °C for 1 min, and a hold step at 72 °C for 5 min, followed by melt curve analysis. The DNA was quantified against the series of standards constructed in-house [20]. The standard series of *sxtA* were generated from DNA isolated from water samples from Harsha Lake using conventional PCR and cloned into Invitrogen’s PCR4 vector using the TOPO TA cloning kit. Each quantification was performed in triplicate on a qPCR plate, which included a six-point standard curve with target gene concentrations ranging from 10^6^ to 10^1^ GCN·μL^−1^ (GCN: genome or gene copy number), using a tenfold serial dilution. PCR inhibition was manually checked by measuring 10-fold diluted DNA extracts using qPCR, and data points where significant PCR inhibition was detected were removed following an established protocol [48].

### 5.4. sxtA Target Sequencing, 16S rRNA Amplicon Sequencing, and Analysis

Each *sxtA* target sequencing, including *sxtA* PCR, library preparation, and sequencing, was conducted as described previously with some modifications [49]. First, primers were designed using gene-specific sequences for the cyanobacteria *sxtA* gene (Appendix A). After that, PCR was performed using 17 μL of Accuprime pfx supermix (Thermofisher, Waltham, MA, USA), 0.5 µL of primers at 10 µM concentration, and 2 μL of prepared DNA. Positive (plasmid clone of *sxtA* isolated from Harsha Lake, described above) and negative (blank water) controls were included. Samples that failed to amplify (as examined by gel electrophoresis) were removed from further analysis. Successfully amplified samples were then cleaned using 14 µL of AMPure XP beads (Beckman Coulter, Brea, CA, USA) with 17 µL of PCR products and eluted in 40 µL of 10 mM Tris (pH 8.5). The cleaned PCR products were normalized at 10 ng/µL, and index PCR was carried out using Accuprime pfx supermix. Further, 17 µL of each PCR product was cleaned with 19 µL of AMPure XP and eluted in 27 µL of 10 mM Tris (pH 8.5). Normalized Libraries (2 nM) were pooled for sequencing using a V3 MiSeq sequencing kit with a 2 × 300, 600 cycle according to the manufacturer’s protocol (Illumina, San Diego, CA, USA).

High-throughput 16S rRNA gene sequencing was also performed to assess changes in microbial communities during the bloom events in the urban lakes in 2021. The sequencing procedure followed the same protocol described for the *sxtA* target sequencing and the primer set targeting the 16S V3-V4 was used.

The raw FASTQ data generated from both sequencings were de-multiplexed using default parameters within the MiSeq Illumina workflow. After removing primers and adaptors, Quantitative Insights Into Microbial Ecology (QIIME2 v2022.02) was used to analyze and de-multiplex paired sequences [50]. Sequences were denoised via the DADA2 plugin, and representative amplicon sequence variants (ASVs) were created with a feature table [51]. The ASVs from 16s rRNA amplicon sequencing were taxonomically assigned using a 16S V4 region-specific classifier based on the SILVA database version 138. For ASVs from *sxtA* target sequencing, MUSCLE and MEGAX [52] were employed to translate and align each ASV, removing those that did not code for the *sxtA* protein or contained stop codons. ASVs that passed quality checks were assessed by using BLAST of NCBI (www.blast.ncbi.nlm.nih.gov/Blast.cgi, accessed on 13 March 2023), and their taxonomic origin was identified.

### 5.5. Data Analysis and Visualization

R version 4.1.3 was used to analyze the final dataset with the help of packages, such as “phyloseq”, “vegan”, “pheatmap”, and “ggplot2” [53,54,55,56]. Different estimates of alpha diversity, such as Chao1 and Shannon Index, were computed, and beta diversity, which denotes the similarities between different bacterial community samples, was analyzed using weighted and unweighted UniFrac metrics. To test for statistically significant differences in community composition and structure between groups, permutational multivariate analysis of variance (PERMANOVA) and non-metric multidimensional scaling (NMDS) were used with the adonis function from the “vegan” package, “ade4”, and “ggplot2” in R. The ‘vegan’ package is recognized for its use in ecological studies and community ecology analyses, particularly through its adonis function for PERMANOVA, which considers the permutation of data points to obtain robust results. NMDS is a dimensionality reduction technique used for visualizing and exploring patterns in multivariate data. Generally, as a rule of thumb, stress values of <0.1 are considered a very good fit, values <0.2 are considered a good fit, and those approaching 0.3 are a poor fit. In addition, eclipses covering 95% of the population were applied to the area of each city. ‘ade4′ provides various methods for multivariate analysis, while ‘ggplot2′ was utilized for data visualizations. Canonical correspondence analysis (CCA) was used to establish the correlation between the abundances of toxin-producing genes and physicochemical water quality parameters via the vegan package, and redundancy analysis (RDA) was employed to explore the relationships between bacterial community structure and the water quality parameters [56].

## Figures and Tables

**Figure 1 toxins-16-00070-f001:**
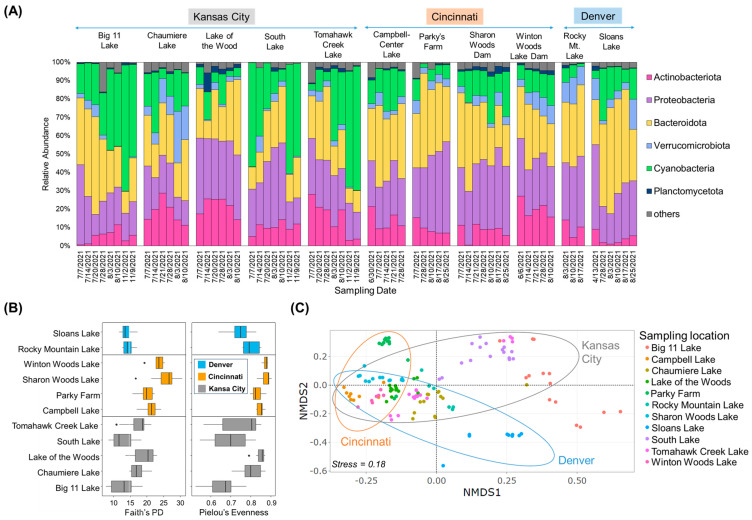
Community composition and structure by 16S rRNA gene amplicon analysis. (**A**) Taxonomic classification (at phylum level) of collected samples from the studied urban lakes, (**B**) Analysis of alpha diversity based on Faith’s PD and Pielou’s Evenness, (**C**) NMDS ordination plot for bacterial community composition (stress = 0.18). Points represent the NMDS scores of each sample in the lakes and ellipses indicate the 95% confidence interval of the group centroids.

**Figure 2 toxins-16-00070-f002:**
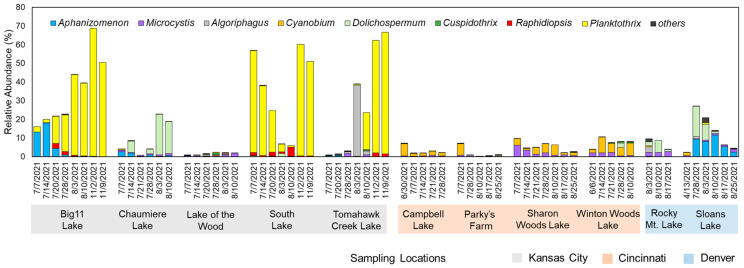
Relative abundances (%) of cyanobacterial genera identified at the studied urban lakes.

**Figure 3 toxins-16-00070-f003:**
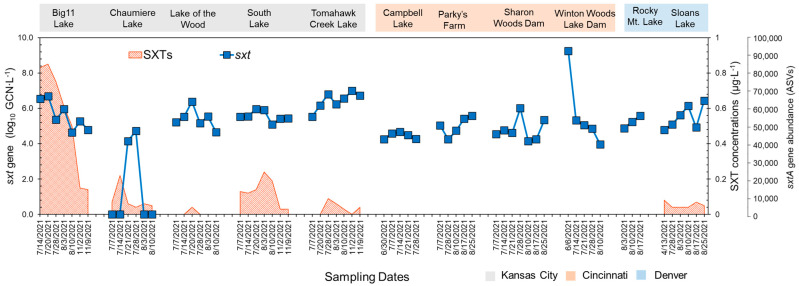
Analysis results of STX concentrations STX concentrations (shaded red area), *sxtA* gene copy numbers by qPCR (blue square), and *sxtA* gene abundances based on *sxtA* target sequencing (red circle) in the lakes of Kansas City, Cincinnati, and Denver. *sxtA* gene abundances were calculated by summing the counts of ASVs identified through the target sequencing. While the primers employed in the qPCR assay for the *sxtA* gene differ from those used in target sequencing, both sets of primers primarily target the *sxtA* gene in *Aphanizomenon* and *Dolichospermum*.

**Figure 4 toxins-16-00070-f004:**
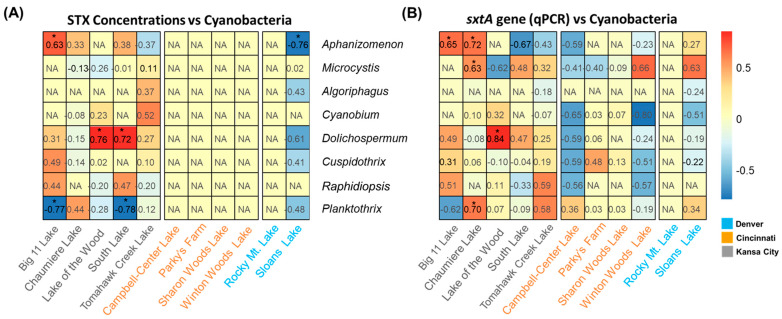
Heatmap showing Pearson correlation coefficients between STX concentrations and abundances of cyanobacteria (**A**), *sxtA* gene (qPCR) and the abundances of cyanobacteria (**B**) in the lakes. NA represents not available. The * indicates 0.01 < *p*-value < 0.05.and South Lake (*R*_pearson_ 0.72, *p*-values < 0.05). *Planktothrix* negatively correlated with the *sxtA* gene and STX in Big 11 Lake (*R*_pearson_ −0.77, *p*-values < 0.05), whereas a positive correlation was observed at Chaumiere lake (*R*_pearson_ 0.70, *p*-values < 0.05). The negative correlation was observed between the relative abundance of *Planktothrix* and the presence of the *sxtA* gene and STXs.

**Figure 5 toxins-16-00070-f005:**
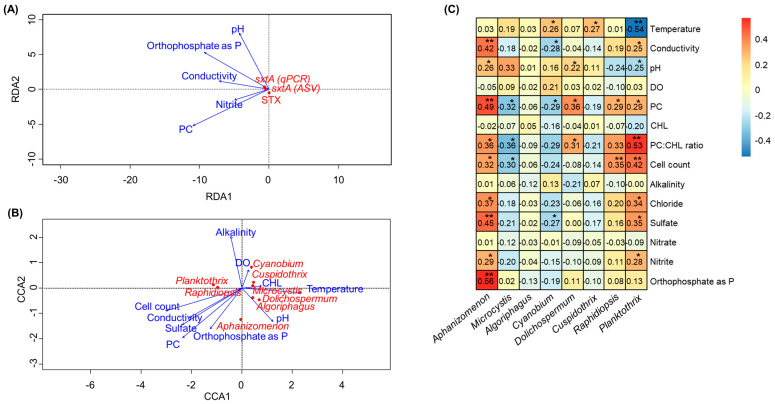
RDA plot linking the qPCR measurements and cyanotoxin concentrations with environmental variables (**A**), CCA plot linking the cyanobacteria species with environment variables (**B**), and heatmap showing Pearson correlation coefficients between the relative abundances of major cyanotoxin producers and environmental variables (**C**). The * and ** represent 0.01 < *p* < 0.05 and 0.0001 < *p* < 0.01, respectively.

**Table 1 toxins-16-00070-t001:** Identified ASVs by BLAST from *sxtA* target sequencing (cut-off: 80% identity, 70% coverage, and 10^−7^ E-value).

Description	Query Cover	E Value	Percent Identity	Accession Number
*Aphanizomenon gracile* NIVA-CYA 676	100%	0	99.49%	LT549449.1
*Aphanizomenon gracile* NIVA-CYA 851	100%	0	99.49%	LT549448.1
*Aphanizomenon gracile* UAM529	100%	0	99.49%	LT549447.1
*Aphanizomenon gracile* NIVA-CYA 655	100%	0	99.49%	LT549446.1
*Aphanizomenon* sp. NH-5 *tatA/E* and *psbH* genes	100%	0	99.49%	EU603710.1
*Dolichospermum circinalis* AWQC131C toxin biosynthesis gene cluster	100%	0	99.49%	DQ787201.1
*Heteroscytonema crispum* UCFS15 strain CAWBG72	99%	4.00E-172	94.64%	MH341392.1
*Heteroscytonema crispum* UCFS10 strain CAWBG524	99%	8.00E-169	94.13%	MH341391.1
*Lyngbya wollei Hypo1*, *Hypo2*, and *Hypo3* genes	100%	8.00E-159	92.41%	EU603711.1

**Table 2 toxins-16-00070-t002:** Pearson correlation testing comparing *sxtA* gene abundance (qPCR) to STX concentrations and *sxtA* gene abundances (ASV counts) in the studied lakes. NA denotes information that is not available. The * indicates *p*-value < 0.05.

Sampling Locations		Correlation Coefficients	
		*sxtA* (qPCR) vs. STX Con.	*sxtA* (qPCR) vs. *sxtA* (ASV Counts)
Kansas City	Big 11 Lake	0.69 *	−0.14
	Chaumiere Lake	−0.13	−0.86
	Lake of Woods	NA	−0.69
	South Lake	0.3	0.69
	Tomahawk Creek Lake	0.36	0.18
Cincinnati	Campbell Lake	NA	0.62
	Parky’s Farm	NA	NA
	Sharon Woods Lake	NA	0.35
	Winton Woods Lake	NA	NA
Denver	Rocky Mt. Lake	NA	NA
	Sloans Lake	−0.61	0.9 *

## Data Availability

All data have been deposited in the NCBI sequence read archive at accession number: PRJNA991726 and PRJNA992779.

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
