# Peer review of "Spatial and Temporal Variability of Saxitoxin-Producing Cyanobacteria in U.S. Urban Lakes"

_toxins, 2024, doi:10.3390/toxins16020070_

Round 1
Reviewer 1 Report
Comments and Suggestions for Authors
Manuscript” Spatial and temporal variability of saxitoxin-producing cyanobacteria in U.S. urban lakes”
The manuscript presents results on seasonal occurrence of the cyanobacterial toxin saxitoxin in different lake in the USA. The study also includes molecular analysis of phytoplankton communities and abundance of cells carrying the sxtA gene (encoding saxitoxin). Knowledge on saxitoxin dynamics with respect to concentrations and organisms producing the toxin in freshwater environments is limited, and therefore the present data constitute a timely relevant contribution to our knowledge gab on occurrence and production of saxitoxin.
A large portion of the presented information is based on bioinformatic analysis. Unfortunately, I am not a bioinformatic expert, and I have slight problems understanding background of some of the conducted analysis, and whether these analyses are the best ones to be applied. I assume several of the readers of articles in TOXINS have the same problem. Perhaps some explanations to the applied tests might be given in supplementary material?
In the study, several correlations between biological and chemical parameter are done, but no final conclusions on processes controlling the production of saxitoxin can be made. As concluded, more studies are needed. I agree to this statement.
I miss some comments on stability or persistence of saxitoxin in freshwater exposed to sunlight. In a recent study (https://www.mdpi.com/2073-4441/14/21/3556), saxitoxin was found to be degraded in sunlight and likewise, the sxtA gene was damaged by natural solar exposure. The present lakes in the USA must have been exposed to sunlight at variable intensities, meaning that light-induced effects probably impacted concentrations of saxitoxin and organisms producing the saxitoxin. Please consider this in the manuscript.
I have a few comments to the manuscript as shown below. I recommend that it is accepted for publication in TOXINS.
Line 14 and 74: Specify that it is free saxitoxin in the water and for example not intracellular saxitoxin.
Line 97-98: Something is missing here: “To evaluate dissimilarities in the overall microbial.”
Line 102-103: In the text “(C) NMDS ordination plot for bacterial community composition (stress = 0.18). Points represent the NMDS scores of each sample in the lakes and ellipses indicate the 95% confidence interval of the group centroids.”, I do not understand what stress = 0.18 is, and there is no mentioning of this analysis (the meaning of the ellipses) in the main text.
Lines 122-126: This text is misplaced!
Figure 3: Good illustration! I might wish more explanation to the differences between qPCR and ASVs of the sxtA gene.
Line 160: What is E-value?
Line 161 and 162: Misplaced text?
Lines 202 and 203: “STX production”: You only measure the amount of saxitoxin being present in the water. You do noy know production rates or degradation rates. Thus, the shown concentrations could, for example, reflect the amount of saxitoxin that was not degraded by sunlight.
- Figure 5: Apparently Aphanizomenon correlated positive with largely all studied parameters. Is this realistic – or does it lead to reflections on whether the applied correlations were the best ones?
- Discussion, first paragraph: Planktothrix was better at competing for light and adjusting to temperatures (although a negative temperature correlation was found, Figure 5). Further, this species had suppressive effects on other algal species. This suggests that Planktothrix might be the winner in the phytoplankton community race. Yet, this was not the case, and this species correlated negatively with chlorophyll. I am not sure I fully understand or agree to your conclusions here.
Line 254: Cylindrospermopsis: The correct genus name is now Raphidiopsis.
Lines 279-280: “Since the major genera of Cyanobacteria identified in the Cincinnati samples were mainly Cyanobium and Microcystis , which are not known to produce STXs, it is plausible that the amplified ASV originated from certain non STX producing species harboring the sxtA gene. In addition, the absence of STXs in the lakes suggest that the gene is either not transcribed into mRNA or serves a different function unrelated to STX production, such as regulating cellular metabolism or synthesizing secondary metabolites.” This is interesting. Could you elaborate a bit more on this/these alternative functions of saxotoxins?
Lines 290-291: “In line with our initial hypothesis, our findings revealed no substantial correlation between STX or the sxtA gene and any individual parameter (Figure 5).” This is intriguing and appears to some extend to be supported by other studies. If there was a degradation of saxitoxin by microbes or damage by sunlight (see the link to DOI above), and if sunlight partly impaired the gene expression, these processes would also have blurred a correlation between sxtA gene abundance and saxitoxin in the water – I think.
Line 341: Weekly sampling performed: Any particular or scientific reason for these weekly intervals? More or less frequent sampling times might have been applied, e.g., to consider bloom episodes.
Lines 359-360: How was cell counting of filamentous species performed? Several of the identified species are filamentous forms, meaning that quantification of individual cells in the filaments are needed. Please explain how cells in the filaments were quantified.
Line 368: ELISA kit: Give sensitivity and concentration range. The shown reference 44 is not useful for researchers who wish to replicate the analysis. Provide a correct reference!
Lines 436 and 437: As a bioinformatic amateur, I have no idea about what “Analysis of Variance (PERMANOVA) and Non-metric multidimensional scaling (NMDS) were used with the adonis function from the “vegan” package , “ade4”, and 437 “ggplot2” in R.” are. You might consider some ways to help future readers of the article!
Reviewer 2 Report
Comments and Suggestions for Authors
The manuscript entitled “Spatial and temporal variability of saxitoxin producing cyanobacteria in U.S. urban lakes” focus on harmful cyanobacterial blooms (HCBs) and the production of saxitoxins (STXs), which pose threats to ecosystems and human health. Using qPCR and sequencing techniques, the research assessed eleven U.S. urban lakes, confirming the presence of the STX-encoding gene sxtA during blooms. The study identified potential STX producers, including Aphanizomenon and Dolichospermum, and compared cyanobacterial abundances with some environmental factors. The topic is relevant and responds to the need to understand cyanobacterial succession and cyanotoxin production beyond the well-studied genus Microcystis and microcystin production. This study is particularly interesting because it found significant concentrations of Stx in several urban lakes, which raises a health safety concern.
Overall, the manuscript is well written and the monitoring of the lakes was well oriented, but important issues arose. Therefore, in my opinion the present manuscript could not be recommended for publication in the journal Toxins before considering the following questions and remarks.
Concern#1: In L88, the authors stated that the relative abundance of cyanobacteria in South Lake peaks in July, but the highest peak occurs in November.
Concern#2: The authors performed a comprehensive comparison with physicochemical parameters of water quality, STX production and cyanobacterial genera. However, the authors did not compare the influence of other microbial communities on the dominance of these genera and toxin production. The influence of some taxa, including Proteobacteria, Bacteroidota and Actinobacteria, on bloom development and decay is widely known. Why did the authors not include these data in the comparative analyses? In fact, they did not find a clear correlation between environmental variables and cyanobacterial taxa. Therefore, the coexisting microbial communities are perhaps the key to explain the observed results. Considering that the main information known to date concerns Microcystis spp. this is even more important, as the authors intended to provide new insights into cyanobacterial genera beyond Microcystis.
Concern#3: In L229-231, the authors explained the dominance of Planktothrix because of it greater competitiveness for light. However, they did not provide any information related to lake turbidity. According to the material and method, the authors used the turbidity sensor on the multiparameter probe, so they should relate this parameter to the presence/absence and abundance of each genus.
Concern#4: The authors also stated that the dominance of Planktothrix could be related to its greater adaptability to a wider temperature range, whereas this trend is not observed in Figure 5.
Concern#5: In L232-234, the authors highlighted that the growth of “Aphanizomenon is inhibited due to the allelopathic effect produced by Microcystis [29]”, and though it has been informed that it occurs in lakes where both species coexist, this trend is not observed in this study. Therefore, the authors should provide a more detailed discussion explaining the results observed in this study.
Concern#6: As mentioned above, the authors used physicochemical parameters to explain their results. They also took into account the competitiveness between the different phytoplankton species. However, they did not consider other, perhaps equally important, biotic interactions, such as bacteria and other predators. For example, algaecidal bacteria are known to play an important role in the succession of filamentous cyanobacteria throughout the bloom period (De Figueiredo et al., 2022; Morón-López et al., 2023). In fact, it has been observed that genera such as Aphanizomenon are more sensitive to algaecidal attack. This could explain the presence or absence of certain genera, and therefore, other factors should be mentioned and discussed.
Morón-López, J., Serwecińska, L., Balcerzak, Ł., Glińska, S., Mankiewicz-boczek, J., 2023. Algicidal bacteria against cyanobacteria: Practical knowledge from laboratory to application. Crit. Rev. Environ. Sci. Technol. 0, 1–28. https://doi.org/10.1080/10643389.2023.2232257
De Figueiredo, D.R., Lopes, A.R., Pereira, M.J., Polónia, A.R.M., Castro, B.B., Gonçalves, F., Gomes, N.C.M., Cleary, D.F.R., 2022. Bacterioplankton Community Shifts during a Spring Bloom of Aphanizomenon gracile and Sphaerospermopsis aphanizomenoides at a Temperate Shallow Lake. Hydrobiology 1, 499–517. https://doi.org/10.3390/HYDROBIOLOGY1040030
On the other hand, minor remarks should be also considered.
1. Please, rewrite L37-38.
2. I think there is a missing sentence in L97-98. The same happened in L 215.
3. Missing bracket in L130.
4. Check terminology, as some Cylindrospermopsis are now Raphidiopsis.
5. In L293, there is a typing mistake in Aphanizomenon.
Reviewer 3 Report
Comments and Suggestions for Authors
The work concerns an important topic, which is the presence of saxitoxin in water reservoirs located near urban areas. The publication is written in a good and understandable language, and the data are presented correctly. However, in my opinion, there are several weak points in the paper. First and foremost, the authors do not specify precisely which types/species in individual lakes are responsible for synthesizing STX. Have such studies been conducted, or were single cyanobacterial cultivations carried out? I believe that microphotographs of the examined cyanobacteria or, better yet, the entire microbiome, would significantly enrich the work. It is not entirely clear to me how the authors determine the concentration of STX. This should be described in more detail in the text, similar to how it was done with PCR. In my view, trends related to STX and the genes responsible for its synthesis, without analyzing specific producers in a particular water reservoir, can only provide a valuable but only a general perspective on the situation related to this issue. Especially since the differences between reservoirs are significant and difficult to interpret without conducting additional precise tests in each of the examined reservoirs. This should be emphasized more strongly in the discussion. I believe that the font on the B and C graphs should be larger. In general, all figures should be submitted in higher quality.
Reviewer 4 Report
Comments and Suggestions for Authors
1. There are many different gene clusters involved in STX biosynthesis in cyanobacteria, so why was only stxA analyzed in this manuscript? The study also found no significant correlation between stxA gene abundances and STX concentrations, so it is not feasible to analyze only stxA?
2. In this manuscript, only molecular methods were used for the identification of cyanobacteria, and there were no results from direct microscopic observations. The results of microscopic observations should be presented.
3. Concentrations of STX in the water were reported in this manuscript, but no detection method was given in the method section.
4. The conclusion section is too extensive and should be condensed.
5. The sampling station maps are suggested to be given in the text.
6. Line 39, “Alexandrium sp” to “Alexandrium spp.”
Reviewer 5 Report
Comments and Suggestions for Authors
This paper reports the spatial and temporal variability of saxitoxin (STX)-producing cyanobacteria in 11 US urban lakes. The samples from Kansas City, Denver and Cincinnati were examined using qPCR of sxtA, one of the STX biosynthetic genes. The water in urban lake is sometimes used as potable water and other activities as the authors write.
The qPCR analysis revealed that the sxtA gene occurred at all lakes, even though STXs were only detected in the lakes from Kansas City and Denver. Big 11 Lake in Kansas City, where the highest concentration of STX was detected, had a signifcant correlation with the abundance of the sxtA gene. It was taxonomically assigned to the genus Aphanizomenon and Dolichospermum. They also showed the results of RDA plot linking the qPCR measurements of sxtA and cyanotoxin concentrations with environmental variables.
Overall, this manuscript provides important information about high correlation of STX and sxtA, and environmental condition of growth of STX producing cyanobacteria, Aphanizomenon and Dolichospermum..
The data shown here is worth to be published in this journal.
Minor points
Page 1 Line 40
I wonder former name Anabaena circinalis is now Dolichospermum circinale, but the authors write Dolichospermum circinalis. I am not quite sure which is correct. Please check it.
In SI, Table S3, Please describe sampling date for each region.
Round 2
Reviewer 2 Report
Comments and Suggestions for Authors
Thank you for considering some of my recommendations.
It is true that the study focus on the producers of STXs, but the authors tried to find some explanation for the dominance and succession of cyanobacterial species during blooms (e.g. see the first paragraph of the discussion). Therefore, in my opinion, although they do not make a deep analysis based on metagenomics or metabolomics, they should mention the importance of other biotic processes, beyond physicochemical factors. Especially if recent research provides data relevant to the discussion of STX-producing species such as Aphanizomenon, Raphidiopsis and Planktothrix.
On the other hand, although the authors agreed that the abundance of cyanobacteria was high in November in South Lake, this change was not included.
Reviewer 4 Report
Comments and Suggestions for Authors
The manuscript has been revised based on the suggestions and is proposed to be accepted for publication.
Author Response
Thank you.